# SimCity: Multi-Agent Urban Development Simulation with Rich Interactions

## Abstract

Large Language Models (LLMs) open new possibilities for constructing realistic and interpretable macroeconomic simulations. We present **SimCity**, a multi-agent framework that leverages LLMs to model an interpretable macroeconomic system with heterogeneous agents and rich interactions. Unlike classical equilibrium models that limit heterogeneity for tractability, or traditional agent-based models (ABMs) that rely on hand-crafted decision rules, SimCity enables flexible, adaptive behavior with transparent natural-language reasoning. Within SimCity, four core agent types (households, firms, a central bank, and a government) deliberate and participate in a frictional labor market, a heterogeneous goods market, and a financial market. Furthermore, a Vision–Language Model (VLM) determines the geographic placement of new firms and renders a mapped virtual city, allowing us to study both macroeconomic regularities and urban expansion dynamics within a unified environment. To evaluate the framework, we compile a checklist of canonical macroeconomic phenomena, including price elasticity of demand, Engel's Law, Okun's Law, the Phillips Curve, and the Beveridge Curve, and show that SimCity naturally reproduces these empirical patterns while remaining robust across simulation runs.

## 1 Introduction

The rapid development of Large Language Models (LLMs), has enabled multi-agent simulations of human societal activities across diverse scales and domains (Gao et al., 2024a). In these settings, autonomous LLM-powered agents interact with each other and with their environment. Existing work has examined not only general social simulacra (Park et al., 2023; Huang et al., 2025; Piao et al., 2025), but also domain-specific applications such as public administration crisis (Xiao et al., 2023), health policy (Hou et al., 2025) or deduction game (Xu et al., 2025). While these work demonstrate the social simulation capabilities of LLM-driven agents, we focus on their application to the **simulation and evaluation of urban-style economic activities**.

For the past two decades, the Dynamic Stochastic General Equilibrium (DSGE) framework has been the predominant paradigm for studying aggregate economic behavior (Blanchard, 2009; Glandon et al., 2023). While mathematically elegant, DSGE models the economy with fully rational agents solving explicit optimization problems in equilibrium, which restricts heterogeneity and richer behavioral dynamics (Sergi, 2018; Vines & Wills, 2020; Storm, 2021). agent-based models (ABMs) offer a bottom-up alternative that accommodates heterogeneity but typically rely on hand-crafted decision rules (Dilaver et al., 2018). Advancing the ABM tradition with recent developments in LLMs, we propose **SimCity**, a multi-agent macroeconomic simulation framework that enables flexible, adaptive behavior with transparent natural language reasoning, and a virtual city environment that provides spatial context.

SimCity models the economy as the interaction of four types of agents: households, firms, a central bank, and a government. To incorporate realistic heterogeneity, households and firms are instantiated as families of agents with rich variation in preferences, abilities and other background, whereas the central bank and government are modeled as single institutional agents, in line with the economic literature (Blanchard, 2025). Agents are implemented through an LLM-based module that integrates environmental observations, agent-specific traits, and structured memory to generate reasoning, planning, and decision-making. Following the classic agent–interaction–environment

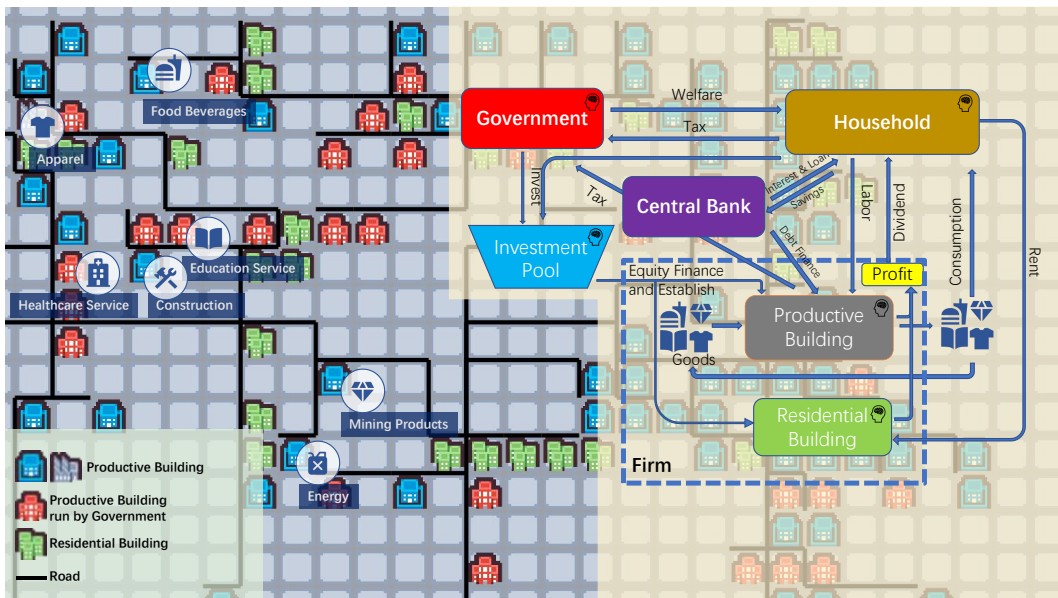

Figure 1: The framework of SimCity. Left: A visualized map with three types of buildings. Right: The rich interactions between various agent modules.

paradigm of multi-agent system design (Wooldridge, 2009), we construct a simulation environment that features a frictional labor market, heterogeneous goods markets, and core financial interactions. To provide spatial context for both agents and researchers, the environment is augmented with a visualized map.

Experimental results show that SimCity successfully exhibits a range of classic macroeconomic phenomena, such as Okun's law, the Phillips curve, the Beveridge curve, price elasticity of demand and Engel's curve. It also simulate the dynamic expansion of a city, and facilitates the simulation of novel economic shocks that were difficult to study in previous frameworks.

In summary, our work makes the following contributions:

1. **LLM-driven macroeconomic simulation with rich interactions**. We integrate LLMs into economic simulation by modeling four distinct economic roles, households, firms, a central bank, and a government, as LLM-driven agents, enabling realistic agent heterogeneity and rich economic interactions.

2. **Visualized urban-style simulation environment**. We provide LLM-based agents with a mapped virtual city that provides spatial context and renders the dynamics of urban expansion, supporting decision-making in a realistic urban setting.

3. **Systematic evaluation**. We compile a comprehensive checklist of canonical macroeconomic phenomena and regularities and use it to demonstrate that our framework exhibits established macroeconomic patterns and generates plausible responses to exogenous shocks.

## 2 RELATED WORK

**LLMs for Social Simulation**    The application of LLMs to social simulation represents an emerging and promising research frontier. LLMs are capable of exhibiting human-like behaviors (Xie et al., 2024) and can be endowed with diverse personas (Chen et al., 2024), offering a basis for interpretable agentic reasoning. Structurally, these simulation frameworks typically contain agents, an environment they interact with, and the interfaces that mediate their interactions (Gao, 2024). Existing work in this domain can be broadly divided into two categories. The first category encompasses general-purpose social simulation frameworks such as  Park et al. (2023), Tian et al. (2025), Tang et al. (2024),  Piao et al. (2025), and  Huang et al. (2025), which aim to model general societal

Table 1: A comparison of agent-driven macroeconomic simulators.

| Simulator | AI Economist | LEN | CATS | EconAgent | Simcity |
|---|---|---|---|---|---|
| Households | 10 | 100 | 100 | 200 | 200 |
| Tax Schedule | Non-linear | $\times$ | $\times$ | Non-Linear | Non-linear |
| Tax Type | Labor | $\times$ | $\times$ | Labor | Labor& Value-Added |
| Agents' Types[a] | HG | HB | HFB | HBG | HFBG |
| Markets[b] | $\times$ | L | LGF | $\times$ | LGF |
| Visualized Map | $\checkmark$ | $\times$ | $\times$ | $\times$ | $\checkmark$ |
| Goods Type | 2 | 1 | 1 | 1 | 10+ |
| Interpretability | $\times$ | $\times$ | $\times$ | $\checkmark$ | $\checkmark$ |

[a]H: Household, F: Firm, B: Bank, G: Government

[b]L: Labor Market, G: Goods Market, F: Financial Market

patterns. The second category investigates specific social phenomena, such as public administration crises (Xiao et al., 2023), health policy (Hou et al., 2025), political manipulations (Touzel et al., 2024), financial market (Gao et al., 2024b), and deduction games (Xu et al., 2025).

**Traditional Macroeconomic Modeling**  DSGE and ABM represent two leading, and in many respects, contrasting approaches to macroeconomic modeling. DSGE features highly idealized decision-makers characterized by precise mathematical objective functions, rational expectations about future variables, and strict market-clearing equilibrium conditions (Stokey et al., 1989; Ljungqvist & Sargent, 2018). In contrast, ABM follows a bottom-up, simulation-driven methodology, where agents are endowed with rule-of-thumb behavior patterns and, in some cases, learning heuristics (Tesfatsion & Judd, 2006; Axtell & Farmer, 2025). Although DSGE remains the dominant approach in economics departments and policy institutions (Woodford, 2009; Negro et al., 2013; Chen et al., 2023), there has been a renewal of interests in ABMs, particularly following the Global Financial Crisis of 2008 and again after the COVID-19 crisis, as mainstream DSGE models struggled to account for these unprecedented shocks (Gatti et al., 2008; Stiglitz, 2018; Borsos et al., 2024). Our work is closer to the ABM tradition. However, we depart from conventional ABMs by replacing predetermined behavioral rules with LLM-powered agents capable of flexible reasoning and information processing in natural languages.

**Comparison to EconAgent**  Li et al. (2024) introduces EconAgent as an early attempt at LLM-driven macroeconomic simulation. Its framework restricts agent-environment interactions to two simplified variables: consumption share and work propensity, which constrains the range of economic activities that can be simulated. Our work extends this approach by incorporating other key economic elements, including firms as LLM-driven agents, heterogeneous goods markets, enriched financial activities and taxation system. Thus, our framework allows the exploration of a wider range of macroeconomic phenomena.

## 3 ENVIRONMENT AND INTERACTION

Our framework consists of three core layers: environment, interaction protocol, and agents. This section describes environment and interaction of our system architecture.

### 3.1 ENVIRONMENT

Each simulation step represents one month. The order of events within each step is detailed in Appendix D.2. By design, agents can only access information from previous steps.

The simulation proceeds in two phases. In phase 1 (*the move-in phase*), new households with synthetic profiles, representing immigrants or newborns, are introduced into the SimCity environment until a predetermined maximum population is reached. Unlike the typical burn-in period used purely for model stabilization in ABMs (Lengnick, 2013), this phase is explicitly designed to study urban expansion, capturing how economic activity and spatial development evolve as the city grows.

In phase 2 (*the development phase*), the population is fixed. However urban developments such as firm creation, investment, and spatial reallocation, continue to take place. We focus on this phase to study macroeconomic regularities, as the system operates in a quasi-steady state that is less susceptible to the transient noise present during the expansion period.

There are three markets in the environment. Firms sell goods to each other and to households in a *goods market*. Notably, goods are qualitatively differentiated (e.g., food, clothing). Firms post jobs and are matched with households in a *labor market*. The central bank accepts deposits and provides loans in a *financial market*. Details of environment setup can be found at Appendix D.1.

We develop a web-based render module to visualize the urban expansion as shown in Figure 1 shown. All buildings are displayed on the rendered map. Geographic placement of new firms are decided by a Visual-Language Model (VLM). For technical details of the module please refer to Appendix D.3.

## 3.2 INTERACTION PROTOCOL

We leverage the common-sense reasoning capabilities of LLMs to act as human-like, heterogeneous agents. Details about the prompts used are provided in Appendix E.1. Agents interacts with the environment by means of *function calling*. The framework loads all the operations that the agent can execute and appends formatted function names along with their descriptions to the prompt. The LLMs will return the actions to be taken and their parameters in JSON format. Framework will execute after a verification. For detailed examples, see the Appendix E.3.

## 4 AGENTS

This section presents an overview of the agents in SimCity. As illustrated in Figure 1, there are four agent types: households, firms, a government, and a central bank, each simulated by an LLM. Building on prior work (Gatti et al., 2011; Wolf et al., 2013; Dawid & Gatti, 2018; Li et al., 2024), our enriched framework expands the decision space considerably.

- Households make four key decisions each period: consumption bundle, labor market action, housing, and financial activity.
- Firms set production levels and prices, decide on hiring and capital investment, and acquire financing when needed.
- The government collects taxes and stimulates the economy through public spending and transfer payments to support consumption and improve social welfare.
- The central bank adjusts the interest rate in response to market conditions.

### 4.1 HOUSEHOLDS

*Households* are the fundamental units in our simulation and interacts with all other agent types and markets. Each household is initialized with a **heterogeneous profile**, including age, education, consumption preference, skill endowment. Details about these characteristics and initialization process can be found in Appendix D.4.

Each month, households receive two types of information: a **personal report** summarizing income, expenditures, and other status changes during the current period, and a **citywide update** containing goods prices, labor market conditions, housing availability, returns on investment and interest rates. Additionally, households are matched with job vacancies created by firms probabilistically.

Based on this information, each household decides on (i) its **consumption bundle**, which requires a minimum expenditure on certain essential goods (Ravn et al., 2008); (ii) its **labor market action** (accepting or rejecting a new job if offered, or resigning from its current job); (iii) its **housing choice**, and (iv) its **financial decisions** (saving, borrowing, or investing in the common investment pool, which is discussed below).

### 4.2 INVESTMENT POOL AND FIRMS

**Investment Pool** An investment pool is implemented as an intermediary module that leverages a VLM for investment decisions. Funds from households' investment actions are temporarily de-

posited into this pool and returned if unused for a month. When the pool accumulates sufficient capital and the VLM deems conditions favorable, a new firm is established.

The pool chooses from a library of 44 **synthetic firm templates**, each of which produces a unique type of good, and selects a a geograhpic placement for the firm. The construction of these templates from real-world data is detailed in Appendix D.5. After a firm is established, all contributing households receive shares of the firm in proportion to their investment.

**Firms**  Each *firm* in SimCity produces a single good. It hires households as workers, invests in capital to improve productivity, and transforms input goods to its specialized output, which is then sold in the market. Firms are instantiated from templates that specify skills requirements of job positions in the firm and the input-output relations governing its production process, where a firm may take possibly more than one input good for its production. The effectiveness of resource utilization (i.e. the ratio at which input goods are transformed into the output good) is determined by a firm-specific Cobb-Douglas production function (Cobb & Douglas, 1928):

$$Y_i = A\,L_i^{1-\alpha}\,K_i^{\alpha}, \tag{1}$$

where $A$ is total factor productivity, $K$ the firm's capital stock, $L$ the effective labor supply, defined as the amount of labor input adjusted for the match between job skill requirements and employees' actual skills.

Similar to households, firms receive **individual reports** summarizing their internal operation (e.g., expenditure composition, employee skill profiles), and **citywide updates** on external conditions such as unemployment rate, interest rates, supply/demand conditions in the goods market.

After deliberating on these information, each firm chooses (i) the **quantity and price** of its outputs, taking into consideration the prices of relevant input goods, as well as the geometric distance to their respective suppliers; (ii) its **labor market actions**, which may include posting job vacancies, laying off employees, or modifying wages; and (iii) **its investment decision**, including whether to borrow from the bank, and whether to purchase fixed assets to increase its capital.

More detailed description of firms can be found at Appendix C.3.

### 4.3  GOVERNMENT

**Macroeconomic Indicators**  The *government* monitors key indicators about the state of the economy, and adjusts taxation and fiscal policies to enhance social welfare, as measured by these indicators. The most important macroeconomic indicators include total consumption, total investment, nominal GDP, and real GDP, where nominal GDP is calculated using current prices, and real GDP uses constant base-year prices. Inflation is tracked in two forms: wage inflation, defined as the rate of change in average wages, and GDP inflation, measured by the rate of change in the GDP deflator. The GDP deflator itself is the ratio of nominal GDP to real GDP.

**Tax and Welfare**  The government collects bracketed income tax from households and value-added tax (VAT) from firms. Let $\mathcal{B} = \{(b_k, r_k)\}_{k=1}^K$ denote a tax schedule, where $b_k$ is the lower threshold of bracket $k$, with $b_{K+1} = \infty$, and $r_k$ the corresponding marginal tax rate. The tax liability from agent $i$ with tax base $z_i$ (income for households, or profit for firms) is:

$$t_i = \sum_{k=1}^{K} r_k \left[ \min\{ z_i,\, b_{k+1} \} - b_k \right]_+, \tag{2}$$

where $[\cdot]_+ \equiv \max\{\cdot, 0\}$. The total tax revenue collected by the government is the summation of taxes from all agents. The government uses this tax revenue in three ways: investing in the construction of public service buildings, which are modeled as government-owned firms, distributing it to households as a universal basic income (UBI), or reserving it.

Mathematical definitions of indicators and other details can be found in Appendix C.4.

### 4.4  CENTRAL BANK AND FINANCIAL SYSTEM

The *central bank* accepts deposits, provides loans, and implements monetary policy. In the first month of each year, it sets the policy interest rate according to a modified Taylor rule, which is a

Table 2: Macroeconomic agent-based modeling checklist.

| Regularity | AI-economist | EconAgent | LEN | CATS | Simcity(Ours) |
|---|---|---|---|---|---|
| Phillips Curve[a] | / | ✓ | × | × | ✓ |
| Okun's Law[b] | / | ✓ | ✓ | ✓ | ✓ |
| Beveridge Curve[c] | / | / | ×[d] | / | ✓ |
| Price Elasticity of Demand[e] | / | / | / | / | ✓ |
| Engel's Law[f] | / | / | / | / | ✓ |
| Investment Volatility[g] | / | / | / | / | ✓ |
| Price Stickiness[h] | / | / | / | / | ✓ |

Legend: ✓: Verified; ×: Verified but inconsistent; /: Mechanism not available in the model.

[a]Short-run inverse relationship between the inflation rate and the unemployment rate.
[b]Inverse relationship between the change in the unemployment rate and the change in real GDP growth.
[c]Job vacancy rate tends to fall when unemployment rises.
[d]The LEN paper claims verified Beveridge Curve, but we are unable to reproduce with https://github.com/newwayland/baseline-economy.
[e]Quantity demanded is downward sloping in price, though the magnitude depends on the type of goods.
[f]As household income rises, food's spending share declines while the absolute amount increases.
[g]Consumption is less volatile than GDP, while investment is more volatile than GDP over the business cycle.
[h]Price has stickiness, i.e., firms don't update price frequently. This is also often called "menu cost".

widely used monetary policy targeting rule (Galí & Gertler, 2007; Dawid & Gatti, 2018). Formally:

$$\hat{r} = \max\big(r^n + \pi^t + \alpha(\pi - \pi^t) + \beta(Y - Y^n), 0\big), \tag{3}$$

where $\hat{r}$ is the policy rate set by the central bank, $r^n$ the long-run natural interest rate, $\pi^t$ the target inflation rate, $\pi$ the GDP inflation in the last period, $Y$ the actual output (GDP), and $Y^n$ the potential output as measured by a linear trend. Parameters $\alpha$ and $\beta$ capture the central bank's responsiveness to inflation and output gap, respectively.

Intuitively, when inflation is high or output exceeds its long-term trend (signals of an overheated economy), the central bank raises interest rate to cool the economy and maintain monetary stability, and vice versa (Galí, 2015). To reflect the gradual adjustment observed in real-world monetary policy, we incorporate a smoothing term consistent with empirical evidence on persistent rate changes (Coibion & Gorodnichenko, 2012).

At each time step, deposits and loans accrue interest: deposit rate equals the policy interest rate, while loan rate equals the policy rate plus a fixed markup. Further details about the financial system are in Appendix C.5.

## 5 EXPERIMENTS

We conduct experiments to evaluate SimCity's ability to perform macroeconomic simulations and to display urban development dynamics. A key limitation of existing frameworks (Lengnick, 2013; Li et al., 2024) is the lack of a common basis for comparison. Prior studies often highlight a relatively small set of selected macroeconomic phenomena, making it difficult to gauge and compare the capacities of different simulation frameworks. To address this gap, we compile a checklist of canonical macroeconomic "stylized facts" from the broad economics literature (Blanchard & Fischer, 1989; Williamson, 2014; Òscar Jordà et al., 2017; Axtell & Farmer, 2025), many of which have previously been used to evaluate other ABM models. We then use this checklist to assess SimCity's ability to reproduce established patterns and to simulate new ones under novel economic shocks. Taking advantage of SimCity's rich framework, we aim to answer the following research questions:

- **RQ1**: What phenomena emerge in SimCity, compared with prior simulation environments?

- **RQ2**: Are emergent regularities robust across multiple simulations?

- **RQ3**: How does SimCity grow during the move-in phase?

- **RQ4**: To what extent can SimCity reflect against external shock?

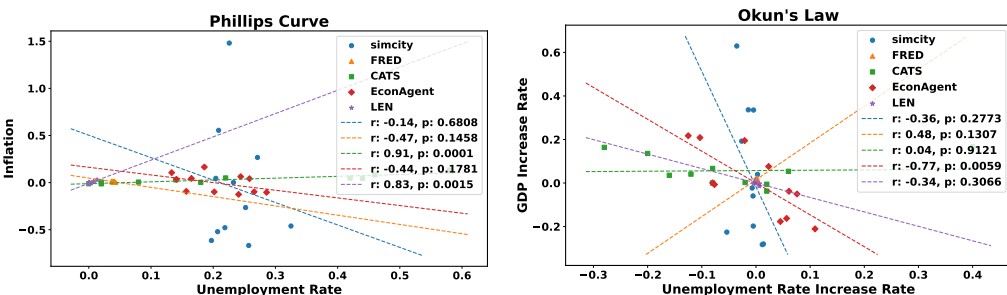

Figure 2: Emergence of the Phillips Curve and Okun's Law in SimCity simulations. $r$-value is the Pearson correlation coefficient and $p$-value indicates the statistical significance of it.

### 5.0.1 EXPERIMENT SETUP

We simulate an economy with a maximum of 200 households. Simulation proceeds in two phases. During the initial phase, which lasts 36 steps (3 years), new households continue to enter. During the second phase, which lasts 144 steps (12 years), the population remains fixed and we observe the simulation under steady-state conditions. We use `gpt-4o-mini` for regular reasoning and `gpt-4` as the vision language model for establishment decision-making. Both are provided by Azure OpenAI API[1] with default sampling parameter. We provide detailed examples of prompts and responses in Appendix E.3.

For comparison, we select two traditional ABMs with predetermined rules LEN (Lengnick, 2013) and CATS (Gatti et al., 2011), a deep multiagent reinforcement learning model AI-economist (Zheng et al., 2022), a LLM-based system EconAgent (Li et al., 2024) and real world data FRED from 1970 Q1 (Federal Reserve Economic Data)[2].

## 5.1 EMERGENCE OF MACROECONOMIC PHENOMENA (RQ1)

A central test of macroeconomic simulations is its ability to exhibit well-documented empirical regularities ("stylized facts") observed in real economies. Leveraging the strong role-play capabilities of LLM agents, SimCity captures many of these regularities that traditional agent-based models have historically struggled to generate. Table 2 summarizes the key phenomena and compares SimCity's performance with baseline models.

**Phillips Curve and Okun's Law**  The Phillips Curve (Phelps, 1967) describes the inverse relationship between unemployment and inflation, while Okun's Law (Okun, 1963) depicts the negative relationship between changes in the unemployment rate and real GDP growth. Following prior work (Li et al., 2024), we use these well-known empirical regularities as major tests of the plausibility of macroeconomic dynamics in our simulation. Recent economic studies using more modern methodologies and granular data document that the Phillips Curve slope is small and has flattened in recent decades (Hazell et al., 2022; Furlanetto & Lepetit, 2024). Consequently, a steeper or more negative slope does not necessarily indicate a more faithful reproduction of the Phillips Curve. As shown in Figure 2, SimCity successfully demonstrates these relationships.[3] We acknowledge that the SimCity points have a large $p$-value (this could be improved by a larger sample size), but we empirically verify that the negative slope can be reproduced with different random seeds in Section 5.2.

**Beveridge Curve**  The Beveridge Curve captures the negative relationship between the job vacancy rate, which is the number of unfilled job openings as a proportion of total job postings, and the unemployment rate (Blanchard & Diamond, 1989; Lengnick, 2013). SimCity successfully exhibits this relationship too, whereas other frameworks in comparison lack the firm module needed to explicitly model job vacancies.

---

[1]https://portal.azure.com/

[2]https://fred.stlouisfed.org/

[3]We omit AI-Economist from this comparison because unemployment is not incorporated in its framework.

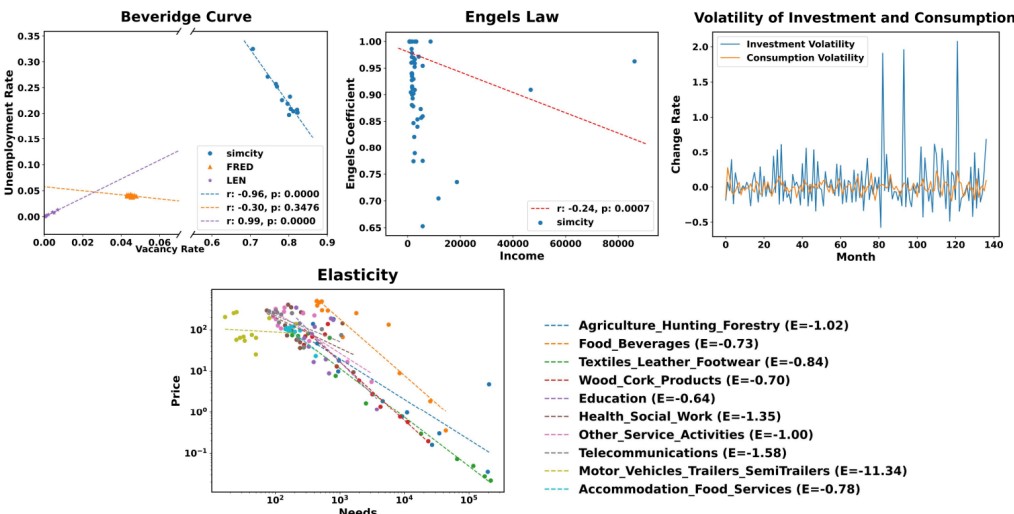

Figure 3: Beveridge Curve and other macroeconomic emergences from SimCity.

**Further Tests** We further validate our model by examining Engel's Law, a foundational empirical regularity, and the Law of Demand, a fundamental economic principle. Engel's Law states that the proportion of income spent on food (the Engel coefficient) decreases as income rises (Chai & Moneta, 2010). Simulations from SimCity correctly exhibit this pattern, reflecting that LLM-driven agents display human-like preferences. Notably, validating Engel's Law requires a heterogeneous goods market, which previous frameworks lacked and therefore could not support.

To test the Law of Demand, we estimate the Price Elasticity of Demand (PED) for each good, defined as $E = \frac{\Delta \log Q}{\Delta \log P}$, which measures how quantity demand responds to changes in price. Standard economic theory predicts a negative elasticity for normal goods, but the magnitude depends on the importance of the good to consumers (Mas-Colell et al., 1995). As shown in Figure 3, SimCity not only exhibits the general negative association predicted by the Law of Demand but also captures the variation in elasticity across goods. For example, necessities like "food/beverages" are more inelastic ($-1 < E < 0$), whereas goods such as "motor/vehicles/trailers/semitrailers" are more elastic ($E < -1$), which is a pattern confirmed by numerous empirical studies (Perloff, 2009; Nelson et al., 2014). Last but not least, as shown in Figure 3 (top-right), our volatility of investment is larger than that of consumption, which matches common-sense.

## 5.2 ROBUSTNESS (RQ2)

An important question for LLM-driven simulations is whether the observed economic phenomena remain stable and reproducible across runs. To address this, we run three sets of experiments with the same hyper-parameters but different random seeds, and the results demonstrate that the observed regularities are robust. Experiment results can be found at Figure 6 in Appendix B.1.

## 5.3 URBAN EXPANSION DURING THE MOVE-IN PHASE (RQ3)

As mentioned in Section 3.1, our simulation involves a phase in which households move in. Figure 4 illustrates the developments during this stage. As shown in the figure, in the first 15 months, GDP steadily increases with the influx of residents. Then, as the influx ceases, the city transitions from a period of expansion to one of stable development after more than ten months of fluctuations. Meanwhile, we note that the VLM, without additional prompts, forms a clustered structure with residential areas near the center and production buildings on the periphery.

## 5.4 CASE STUDY: EXOGENOUS PRICE IMPULSE (RQ4)

We separately apply a price-down and a price-up impact on the city. In the beginning of year 15, we randomly select 7 of 44 goods in two different simulations and apply a 50% price-down and a price-up impulse separately. We then let the simulation run for 6 years. Price-down impulse response is

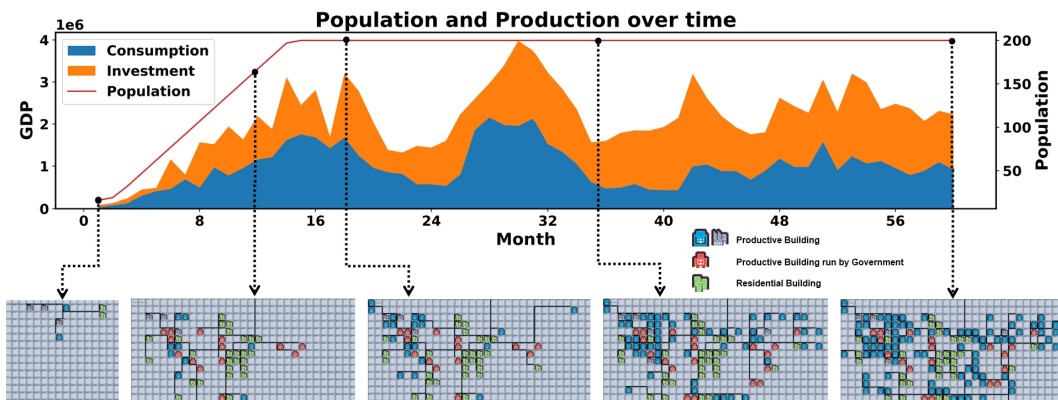

Figure 4: GDP & population curves, and map changes during the move-in phase.

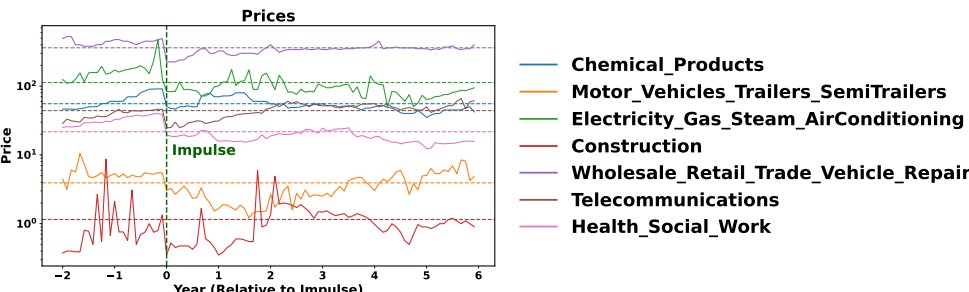

Figure 5: External impulse (at year 15) does not significantly affect tendency of prices of goods.

shown as Figure 5 and price-up at Figure 7 (Appendix B.2). The experiment shows that the good's price level gradually returns to equilibrium after an exogenous shock. Due to the effect of price stickiness, the response to the exogenous shock will take some time. Note that dotted lines in the figure represent long-run average prices of goods. Since the shock is one time and the fundamentals of the economy did not change, prices revert close to their original levels in the long-run.

## 6 LIMITATIONS

**Abnormal Behavior from Agents.** Our agent's decision-making is powered by a LLM. A significant challenge arises from the inherent difficulty in imposing robust constraints on the LLMs' decisions. Furthermore, there exists a non-trivial probability that the LLM may generate highly aberrant or unexpected decisions, such as altering a price by several orders of magnitude. While the likelihood of any single such event is small, its probability becomes non-negligible over hundreds of iterations. In this work we apply simple heuristic-based checks prevent those egregious behavior.

**Lack of Complex Financial Activities** Our current simulation framework does not incorporate real-world financial markets, such as those for bonds, stocks, or their derivatives. Moreover, the simulated agents do not employ mathematical optimization techniques in their investing strategies. These simplifications may limit the model's ability to accurately simulate complex market dynamics.

## 7 CONCLUSION

With SimCity, we conduct complex simulations that incorporates four distinct LLM-driven economic roles (households, firms, the central bank and the government), with a visualized map and markets with heterogeneous goods. Experiments show that SimCity robustly matches a checklist of established macroeconomic phenomena and exhibit plausible responses to exogenous shocks. We hope SimCity could serve as a solid base platform towards more a realistic economic simulation.

# 8 ETHICS STATEMENT

This work utilize LLM-driven agents to simulate. No sensitive or private data is used. Names of agents are generated by LLMs and are not intended to represent any real individuals or entities. Our simulation has not been quantitatively calibrated with real-world data and does not constitute a prediction of reality.

# 9 REPRODUCIBILITY STATEMENT

To help with the reproducibility of our work, we provide all LLMs prompts in Appendix E.1. We call LLMs API with default sampling parameters. We repeat experiments with different random seeds (Appendix B.1) to ensure that our main results are robust. We plan to release code in near future.

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

## A  STATEMENT ON LLM USAGE

This work utilize LLM-driven agents to simulate. In addition, we use LLMs to improve the grammar and readability of this manuscript. The LLM was not used for any other scientific aspects of this work, and all intellectual content is solely the product of the authors.

## B  EXPERIMENT DETAIL AND ADDITIONAL RESULTS

**LLMs Costs**  Our simulation costs about 800,000 tokens, which is roughly $0.25 per step. For a standard 180 steps simulation, the total cost is around $180.

### B.1  ROBUSTNESS

We run three sets of experiments with the same hyper-parameters but different random seeds, and the results demonstrate that the observed regularity is reproducible.

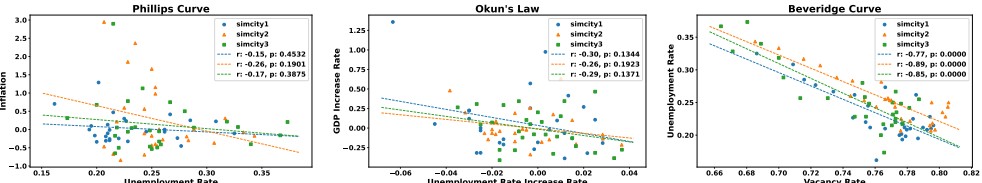

Figure 6: The results from different random seeds demonstrate that the observed regularity is robust.

### B.2  PRICE IMPULSE

The price-up impulse response, Figure 7, is shown here due to space limitation.

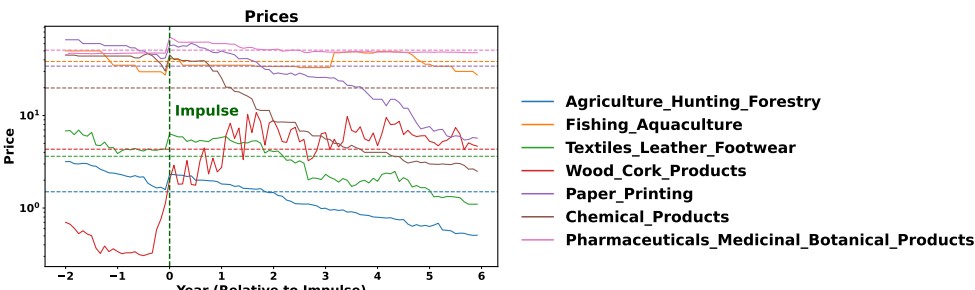

Figure 7: External impulse does not significantly affect long-run prices.

## C  AGENTS DETAILS

### C.1  NOTATIONS

We define the following notations:

- The goods set, $\mathcal{G}$, where $|\mathcal{G}|$ denotes the number of heterogeneous goods present in our simulation. Certain subsets of $\mathcal{G}$ have a specific meaning, such as $\mathcal{G}_E$ for essential goods, $\mathcal{G}_T$ for transportation goods and $\mathcal{G}_D$ for durable capital.
- Each step in our simulation represents a month. For clarity, time subscripts are suppressed unless needed for exposition. For example, the labor choice at time $t$ for household $i$ is written as $l_i$ instead of $l_{i,t}$.

## C.2 HOUSEHOLDS

A household $i$ is characterized by the following attributes:

- **Cash Holdings** $m_i$: The amount of money held by the household.
- **Labor Participation:** An indicator $l_i \in \{0, 1\}$, where 1 means the household is employed to work in a position denoted by $\text{position}_j$ and receives a salary $S_j$, and 0 means the household is unemployed. Note we are modeling the extensive margin of labor, instead of the intensive margin (hours worked).
- **Innate Skills:** Each household is endowed with heterogeneous skills. Please see Table 3 for the full list of skills. For each skill $j$, the innate skill level $s_{i,j} \sim U(s_{\min}, s_{\max})$ means that when the household works in a job requiring skill $j$, it supplies $l_i \times s_{i,j}$ effective units of labor. Additionally, the probability that a household is matched with a vacancy requiring skill $j$ is an increasing function of $s_{i,j}$. This is in line with the directed search literature in economics (Wright et al., 2021).
- **Heterogeneous Consumption Preferences:** The household has needs denoted by $n_i$. For each good $g \in \mathcal{G}$, $n_{i,g}$ represents the desired number of units of good $g$ to be purchased. $n_i$ may split to essential needs $n_{e,i}$ which is fixed denoting immutable demands and additional needs $n_{a,i}$. The modeling of essential goods has a long history in economics. See for example, the Stone-Geary Utility function (Stone, 1954).
- **Housing:** Each household resides in a house $\mathcal{H}_i$. Consequently, the household must pay rent $r_i$ to the owner of the property[4].

## C.3 FIRMS

Each firm $i$ is characterized by the following attributes:

- **Cash Holdings** $m_i$: The monetary resources currently available to the firm.
- **Output Good:** Each firm specifies a specific type of good $g_i \in \mathcal{G}$.
- **Shareholders:** A collection of households and/or government entities, each holding a share denoted by $\textbf{share}_{i,j}$.
- **Dividend Rate** $d_i$: A parameter $d_i \in [0, 1]$ representing the fraction of the firm's profits that are distributed as dividends to shareholders. Specifically, a shareholder holding $\textbf{share}_{i,j}$ receives a dividend computed as:

$$D_j = \frac{\textbf{share}_{i,j} \, d_i \, m_i}{\sum \textbf{share}_i}.$$

- **Job Positions** $\mathcal{J}_i$: A list of job vacancies within the firm. Each position $j$ requires a specific skill and offers a salary $S_{i,j}$.
- **Durable Capital** $\mathcal{K}_i$: The stock of durable capital assets that the firm has invested in to support production.
- **Estimated Value** $\mathcal{V}_i$: The firm's estimated value, calculated as:

$$\mathcal{V}_i = \frac{\sum_{j=t-12}^{t-1} \text{profit}_j}{12 \cdot \max(\mathcal{I}_{d,t}, \epsilon_d)} + m_i - L_i + \mathcal{K}_i.$$

where $\mathcal{I}_{d,t}$ denotes the GDP deflator, and $\epsilon_d = 0.02$ is a lower bound introduced to prevent division by zero.

- **Loans**: The amount of loan the firm owes to the financial system.

If a firm experiences zero income for a year and its assets are insufficient to cover its overdue debts, the firm is declared bankrupt. In that case, all employees are terminated, and the land it occupied is released.

A rentable apartment (residential building) is also formally modeled as a firm, though its decision-making is limited to adjusting its rental price based on conditions in the housing market.

---

[4]In this model, we assume that everyone rents their housing. In reality, if a household owns its home, one can think of it as renting a property in which it holds full ownership.

### C.4 GOVERNMENT

The government records a series of indicators:

- **nominalGDP**$_t$: the sum of consumption, investment, and government spending, and **realGDP**$_t$, computed things above using the base year's prices.
- **eq**$_t$, **giniW**$_t$, **giniI**$_t$: measures of equality, wealth inequality (Gini coefficient), and income inequality (Gini coefficient), defined as follows. Suppose there are $n$ households, and let $\{x_{(i)}\}_{i=1}^n$ denote the sorted (in increasing order) wealth (or income) of households. Then,

$$\mathbf{giniW}_t = \frac{\sum_i \text{sorted\_wealth}_i \cdot 2i}{n \cdot \text{sum\_wealth} - \frac{n+1}{n}}, \tag{4}$$

$$\mathbf{giniI}_t = \frac{\sum_i \text{sorted\_income}_i \cdot 2i}{n \cdot \text{sum\_income} - \frac{n+1}{n}}, \tag{5}$$

$$\mathbf{eq}_t = 1 - \frac{n}{n-1}\mathbf{giniI}_t. \tag{6}$$

- Unemployment rate $u_t$, along with broad monetary aggregates M0 and M1[5].
- Goods production $Y_t$, and average production $\bar{Y} = \frac{\sum_t Y_t}{T}$.
- Inflation: including wage inflation $\mathcal{I}_w$ and GDP inflation $\mathcal{I}_d$, each defined by:

$$\mathcal{I}_{w,t} = \frac{\sum_{\text{household}} S_{i,t}}{\sum_{\text{household}} S_{i,t-1}} - 1, \tag{7}$$

$$\mathbf{GDPdeflator}_t = \frac{\mathbf{nominalGDP}_t}{\mathbf{realGDP}_t}, \tag{8}$$

$$\mathcal{I}_{d,t} = \frac{\mathbf{GDPdeflator}_t}{\mathbf{GDPdeflator}_{t-1}} - 1. \tag{9}$$

### C.5 CENTRAL BANK AND FINANCIAL SYSTEM

We implement a Taylor rule with interest rate smoothing in the policy behavior of our simulated central bank agent. Without smoothing, simulated interest rates exhibit implausibly high volatility in response to small shocks.

Interest rate smoothing refers to the empirical regularity that central banks adjust policy rates gradually over time, rather than immediately to the level implied by contemporaneous macroeconomic conditions. This behavior is often interpreted as reflecting forward-looking policy preferences, model uncertainty, or an aversion to financial market instability. In practice, this means that the actual interest rate set by the central bank is

$$r_t = \rho r_{t-1} + (1-\rho)\hat{r}_t, \tag{10}$$

where $\hat{r}$ is the policy rate computed from the unmodified Taylor rule 3, $r_{t-1}$ the interest rate set last period, and $\rho$ is the smoothing factor.

Empirically, interest rate smoothing is a well-documented phenomenon. For example, Coibion & Gorodnichenko (2012) estimate a Taylor rule with smoothing and find that the smoothing coefficient typically ranges between 0.7 and 0.9 in developed economies. We choose $\rho = 0.8$ as our parameterization.

## D FRAMEWORK DETAILS

### D.1 ENVIRONMENT SETUP

There are 44 types of goods in SimCity. Each of them represents of an economic industry in the real world as categorized by Organization for Economic Co-operation and Development (OCED).

---

[5]M2 is not included since fixed deposits have not been defined.

Initial prices of all goods are set at 50. However, the specific choice of initial prices do not matter for our simulation. We experimented with various initial prices but in all cases, as we mention in 5.4, prices reach steady-state in a small number of steps. Similarly, wages of positions and rents of residential buildings are set arbitrarily.

## D.2 STAGES WITHIN EACH STEP

As aforementioned, each simulation step represents one month, and each step includes the following four stages:

- **Production and Trading Stage:** Firms produce goods, after which households and firms purchase goods for consumption and materials as planned.
- **Taxation and Dividend Stage:** Firms pay dividends, while the government collects taxes and disburses welfare.
- **Metabolic Stage:** New companies are established through equity financing from the investment pool, bankrupted companies are removed, and the population grows according to set rules.
- **Revision Stage:** Households, firms, the government, and the central bank agents review their situations and decide on their actions. Only the revision stage involves LLM agents.

## D.3 RENDER MODULE

We utilize `flask-socketio` to build a web-server and `Vue.js` to build a website.

The assets are from the open-source `tiny-battle` package created by Kenney[6].

## D.4 PROFILE SETUP

Age distribution is from the Demographic and Housing Characteristics (DHC) table from U.S. Census Bureau[7].

To assign realistic initial cash holding to households, we estimate the income distribution using U.S. microdata. We use the 2023 American Community Survey (ACS) IPUMS microdata, which contains detailed household-level income information. We assume a lognormal distribution for household income, a standard approximation in the economics literature due to its simplicity and its ability to capture the right-skewed nature of income data.[8] Formally, the initial cash holding is drawn from

$$\ln m_{i,0} \sim \mathcal{N}(\mu, \sigma^2) \tag{11}$$

where $\mu = 11.1496, \sigma^2 = 1.1455$, which are estimated via maximum likelihood estimation (MLE).

## D.5 SYNTHESIS

We assume that each category of goods is produced by a representative type of firm. Therefore, we generate a firm template for each category, which specifies the firm's name, the skill requirements for its positions, and an input-output "recipe" derived from the Input-Output Tables (IOTable) provided by the OECD.[9]

We process the IOTable to determine the inputs for each firm. For each category of goods, we normalize the input requirements from all other categories needed to produce one unit of output. We then select the top input categories that cumulatively account for more than 75% of the total input value, defining these as the necessary inputs for the corresponding firm's production process.

---

[6]`www.kenney.nl`

[7]`https://data.census.gov/table?q=PCT12&d=DEC+Demographic+and+Housing+Characteristics`

[8]While some heavier-tailed distributions (e.g. Pareto) better approximate top incomes, our simulation does not focus on the super-rich. Hence, the lognormal distribution is a reasonable choice. See Hlasny (2021) for a recent summary of related work.

[9]`https://www.oecd.org/en/data/datasets/input-output-tables.html`

Then, we use the prompt introduced in Listing 1 to call a LLM to generate the processed input-output into a firm template.

Observing that the positions required skills in the generated firm templates contain many semantically similar skills, we use the prompt introduced in Listing 2 to call an LLM to merge these similar skills. Skills after merge is shown at Table 3.

Listing 1: Prompt for firm template generation.

```
I am currently developing a game. One of the tasks I need to do now is:
    Given a recipe, please help me design a building to produce this
    recipe.
First, I will give you a recipe, which is composed of some raw materials.
    Then, you need to design a suitable building to produce it. Noticing
    that the employees' core skill should be as abstract as possible.

``type`` must either "BusinessBuilding" or "ServiceBuilding". If you
    consider a type of product is mainly provided by government, you
    should set it as ServiceBuilding, else BusinessBuilding.
Here is an example:

```
### Recipe ###

{
  "input": {
    "RawFood": 1000
  },
  "output": {
    "Food": 1000
  }
}

Your Response:

  {
    "template_name": "Restaurant",
    "width": 1,
    "height": 1,
    "description": "A restaurant that uses raw food to serve food.",
    "type": "BusinessBuilding",
    "building_cost": 500000,
    "provide_radius": 10,
    "employees": [
      {
        "position_type": "Manager",
        "salary": 3000.0,
        "core_skills": [
          "Management"
        ],
        "importance": 1.0
      },
      {
        "position_type": "Cook",
        "salary": 1000.0,
        "core_skills": [
          "Cooking"
        ],
        "importance": 0.5
      },
      {
        "position_type": "Waiter",
        "salary": 1000.0,
        "core_skills": [
```

```
      "Service"
    ],
    "importance": 0.5
  },
  {
    "position_type": "Waiter",
    "salary": 1000.0,
    "core_skills": [
      "Service"
    ],
    "importance": 0.5
  }
],
"recipes": [
  {
    "input": {
      "RawFood": 1000
    },
    "output": {
      "Food": 1000
    }
  }
]
}
```

Now the recipe I provide will listed follow "### Recipe ###"
You should respond with a factory design in the same format as the
    examples above.

Listing 2: Prompt for merging skills

```
[Skills]
```
The content above is a list of several skills from my economic simulation
    environment, SimCity. A job posting might require several of these
    skills. I now feel that there are too many of them and they need to
    be simplified. Please merge the skills that have the same meaning
    into a single skill (note: only merge two skills if and only if they
    are very similar; for example, Geology and Mining should not be
    merged because they are different specialties), and provide the merge
    relationships: a JSON object { "A": "B" } indicates that skill A is
    essentially skill B.

If several skills are to be merged into one category, choose the most
    representative one as B. Then, return this JSON object.

Table 3: List of skills.

| | | | |
|---|---|---|---|
| Management | Quality Control | Operations Management | Logistics Management |
| Physical Labor | Aquaculture | Environmental Science | Engineering |
| Monitoring | Maintenance | Geology | Mining |
| Chemistry | Surveying | Technical Skills | Food Science |
| Machinery Operation | Safety Management | Supply Chain Management | Pharmaceutical Science |
| Laboratory Skills | Metalworking | Assembly | Energy Management |
| Water Management | Equipment Maintenance | Project Management | Design |
| Building | Sales | Customer Service | Vehicle Maintenance |
| Data Analysis | Regulations | Equipment Operation | Driving |
| Transportation | Communication | Culinary | Cleaning |
| Media Production | Equipment Handling | Logistics | Information Technology |
| Finance | Technical Support | Real Estate | Insurance |
| Market Analysis | Office Management | Administrative Support | Building Maintenance |
| Research | Consulting | Facility Maintenance | Basic Repairs |
| Support | Teaching | Marketing | Legal Knowledge |
| Human Resources | Sanitation | | |

# E   PROMPTING AND EXAMPLES

---

**Structure of Prompt**

**System Prompt:**

You are a citizen of SimCity and you are taking action improve your life. Basically speaking, your goal is to be consuming a greater variety of products, acquiring more money and assets, and having a stable job and residence.
There are some information about you and the city which may affect your life . . .

**User Prompt:**

User Prompt:
### Profile
Your name is [name], and you are [age] years old . . . . . . You current have [money] money. You are working at [place] as [position] with salary [salary] . . .
### Report
. . . During the past 12 months, your average income is [average_income]. This month, your income is [total_income], which consists of the following parts :
 - Salary : . . .
 - Benefit : . . .
. . .
This month, your outcome is [total_outcome]
. . .
### Observation
Here are opening positions in the city :
 -[position_id]: . . .
. . .
The social average return on investment is [roi], and the interest rate of the bank is [interest_rate].

Figure 8: Structure of Prompt

---

## E.1   PROMPTING STRUCTURE

We aim to leverage the common-sense capabilities of large language models to act as human-like, heterogeneous agents. Figure 8 provides the structure of the prompt for a household agent.

As mentioned in 4, the Agent will receive necessary information to make its decisions. The Profile includes some basic personal information about the Agent, such as current income, age, skills, job, and residence. The Report includes some of the Agent's recent experiences and information. The Observation presents information about the overall environment that the Agent should be aware of,

Table 4: Example Listings

| Agent Type | System Prompt | User Prompt | Response |
|---|---|---|---|
| Household | Listing 3 | Listing 4 | Listing 5 |
| Government | Listing 6 | Listing 7 | Listing 8 |
| Investment Pool | Listing 9 | Listing 10 | Listing 11 |
| Firm(Productive Building) | Listing 12 | Listing 13 | Listing 14 |
| Firm(Residential Building) | Listing 15 | Listing 16 | Listing 17 |

such as vacant positions, the unemployment rate, prices of goods and the Return on Investment (ROI).

Prompts for other agents are structured in a similar manner. Detailed examples are provided in Appendix E.3.

### E.2 FUNCTION CALLING

The model interacts with the environment by means of Function Calling. The framework loads all the operations that the Agent can execute and appends formatted function names along with their descriptions to the prompt. The LLMs will return the actions to be taken and their parameters in JSON format. Framework will execute after a verification.

### E.3 FULL EXAMPLES

Full examples of all types of agents can be found at Table 4.

Listing 3: System prompt example of household agent

```
You are a citizen of SimCity and you are taking action improve your life.
    Basically speaking, your goal is to be consuming a greater variety
    of products, acquiring more money and assets, and having a stable job
    and residence.
There are some information about you and the city which may affect your
    life. The information including four parts: "### Profile", "###
    Report", "### Observation", and "### Actions".
You will be provided your personality traits and skills, and some
    personal information, which will be listed after "### Profile".
You have also provided your last month's living status, which will be
    listed after "### Report".'
Any information about the city which may affect your life will be listed
    after "### Observation".
Based on this information, you may take some actions to improve your life
    . These actions you may take will be listed after "### Actions".

You should only respond in JSON format as described below:
{
    "reasoning": "reasoning"
    [
        "action1(para1, para1, ...)",
        "action2(para1, para1, ...)",
        ...
    ]
}

Ensure the response can be parsed by Python `json.loads`, e.g.: no
    trailing commas, no single quotes, etc.
```

Listing 4: User prompt example of household agent

```
### Profile
```

```
Your name is Donald, and you are 61 years old.You current have 2667.408
    money in your account.
You are a Active,Unadventurous man
You are working at Mining Facility as Engineer with salary 2200. This
    work requires skills: Engineering.

You are living in Residential Building at (12, 11) with rent 900. Your
    work place is at (15, 19). The distance between your home and your
    work place is 11.

You have experienced the following education:
 - K-12
 - Bachelor
 - Master

You plan to spend a percentage of your income on the following items:
 - Agriculture_Hunting_Forestry: 0.1
 - Fishing_Aquaculture: 2.0
[... and other needs.]
 - Household_Employer_Activities: 0.0
 - Motor_Vehicles_Trailers_SemiTrailers: 0.0

Bank Account Donald:
You have no savings
You have no loans.
Demand deposit interest rate is 56.0464%, and loan rate is 58.0693%.
Your loanable amount is 677334.81606351.

### Report

This month, your income is 5467.408, which consists of the following
    parts:
 - Salary: 2800.000(51.2126%)
 - Welfare: 2667.408(48.7874%)

This month, your outcome is 3753.360, which consists of the following
    parts:
 - Consumption: 2878.360(76.6876%)
   - Agriculture_Hunting_Forestry: 290.547(10.0942%)
   - Wood_Cork_Products: 560.000(19.4555%)
   - Education: 280.000(9.7278%)
   - Other_Service_Activities: 1400.000(48.6388%)
   - Electricity_Gas_Steam_AirConditioning: 280.000(9.7278%)
   - Food_Beverages: 67.813(2.3560%)
 - Tax: 0.000(0.0000%)
 - Rent: 875.000(23.3124%)

### Observation

Here are opening positions in the city:
 - position_id: 2000149, Warehouse Worker requires skills Physical Labor,
     Logistics Management, with salary 350
 - position_id: 2000330, Maintenance Worker requires skills Maintenance,
     Physical Labor, with salary 500
[... and other available positions.]
You expect the work should cover your spending of 3753.360.

Here are available residential buildings in the city:
 - residential id: 200006 at (11, 1) with rent 875
 - residential id: 200051 at (7, 14) with rent 875
[... and other residential buildings' position and rent]
 - residential id: 200036 at (12, 5) with rent 900
 - residential id: 200052 at (21, 7) with rent 875
```

The social average return on investment is 13.1554%, and the interest
    rate of the bank is 56.0464%.
If roi is higher than the interest rate of the bank, you can invest your
    money in the investment market to get a higher return on investment
    and you may borrow money from the bank to invest if you need.
Otherwise, you may simply save your money in the bank.

The prices of goods on the market are as follows:
  - Agriculture_Hunting_Forestry: 121.652, 35.9921%
  - Fishing_Aquaculture: inf, 0.0000%
  - Food_Beverages: 137.423, 35.5278%
[... and other prices and their tendency. ]
  - Household_Employer_Activities: inf, 0.0000%
  - Motor_Vehicles_Trailers_SemiTrailers: 75.000, 130.7692%
When adjusting your needs for goods, please keep the following points in
    mind:
1. You can adjust your needs for goods based on the price change rate.
    All change rates are derived from the average prices of the past 12
    months.
2. Your demand for goods should also reflect your income. If your income
    increases, you should allocate a higher percentage of your income to
    luxury items and a lower percentage to food.
3. If your income decreases, you should allocate a lower percentage of
    your income to luxury items and a higher percentage to food.

During the past 12 months, your average income is 733.333:
- 12 months ago, you were unemployed, and your income was 0.000.
- 11 months ago, you were working at Maintenance Worker, and your income
    was 0.000.
- 10 months ago, you were unemployed, and your income was 0.000.
- 9 months ago, you were working at Laborer, and your income was
    1500.000.
- 8 months ago, you were working at Laborer, and your income was
    1300.000.
- 7 months ago, you were working at Laborer, and your income was 0.000.
- 6 months ago, you were working at Production Worker, and your income
    was 2000.000.
- 5 months ago, you were working at Warehouse Worker, and your income was
     0.000.
- 4 months ago, you were working at Production Worker, and your income
    was 1200.000.
- 3 months ago, you were working at Warehouse Worker, and your income was
     0.000.
- 2 months ago, you were working at Engineer, and your income was 0.000.
- 1 months ago, you were working at Engineer, and your income was
    2800.000.

### Interfaces
Here are the actions you may take:
  - resign(): If you are currently employed, you will be fired from your
      current job.
  - find_job(position_id: int): Choose a job. If you are already employed,
       you will be fired from the current job. You may choose not to work
      even if you are unemployed to wait for a better opportunity.
  - modify_needs_percentage(name: str, percentage: float): Set your needs
      of a specific item to a certain percentage of your income. 1 means
      100% of your income, 0.5 means 50% of your income. Higher and more
      varied needs may lead to higher happiness, while lower needs may
      save your money.
  - set_invest_rate(rate: float): Set the investment rate. The investment
      rate is the percentage of your money that you will invest into the
      investment market each month. It should be between 0 and 1.

```
- borrow(amount: float): Borrow money from the bank, those money will be
    paid back with interest. You may borrow money from the bank for
    investment or to purchase necessary consumable goods.
- payback(loan_id: int): Pay back the loan with the given loan_id.
- save_money(amount: float): Save money to the bank to get interest.
- withdraw(amount: float): Withdraw money from the bank.
- move_to_home(building_id: int): Move your home to a specific building.
    If your are currently homeless, you will move into the building.
- wait(): Your life is pretty good now. What you want is to keep the
    status quo.
```

Listing 5: Response example of household agent

```
{
 'reasoning': "At 61 years old, it's crucial to manage your finances and
    job stability. Given your current job as an Engineer at a Mining
    Facility with an income of 2800, your expenses are relatively high
    at 3753.360 monthly. You need to either adjust your spending habits,
     take a second job with potentially lower salary but manageable
    workload, or invest wisely. Your current plan shows significant
    consumption in various sectors. Considering the job market, there's
    an opportunity as a Metallurgist, which aligns with your Engineering
     skills and offers a higher salary. Resigning from your current job
    to pursue this position could help improve your finances
    significantly. Additionally, to prepare for a potential career
    transition, you might consider modifying your spending to save more
    .",
 'actions':
  [
   'resign()',
   'find_job(1000008)', # 1000008 here is a job id
   "modify_needs_percentage('Food_Beverages', 0.15)",
   "modify_needs_percentage('Other_Service_Activities', 0.3)",
   "modify_needs_percentage('Electricity_Gas_Steam_AirConditioning', 0.1)
      "
  ]
}
```

Listing 6: System prompt example of government agent

```
You are the governing body of SimCity, entrusted with the task of making
    strategic decisions to improve the city's overall standard of living.
There are four primary categories of information available to you that
    will affect your decision-making process: "### City Profile", "###
    City Statistics", "### Citizen Feedback", and "### Potential
    Initiatives".

The "### Report" section contains data regarding the city's current
    economic, social, and environmental status..

You should respond in JSON format as described below:
```
{
    "strategies": "strategies",
    [
        "initiative1(para1, para1, ...)",
        "initiative2(para1, para1, ...)",
        ...
    ]
}
```
Ensure the response can be parsed by Python `json.loads`, e.g.: no
    trailing commas, no single quotes, etc.

An example output might look like this:
```

```
```
{
    "strategies": "Given the city's high unemployment rate and the
        citizens' feedback about lack of job opportunities, it would be
        beneficial to invest in job creation programs. Furthermore, the
        high levels of pollution suggest a need for environmental
        initiatives.",
    "actions": [
        "LaunchJobCreationProgram('Technology Sector')",
        "ImplementEnvironmentalPolicy('Recycling Initiative')"
    ]
}
```

Listing 7: User prompt example of government agent

```
### Report

Here is the government report:
The government has a balance of 1.102307578548789e-09.

Bank Account Government:
You have no savings
Loans:
  - id: 369, loan amount: 13924261.356747806, should be paid back at
    1973-01

Demand deposit interest rate is 56.0464%, and loan rate is 58.0693%.
Your loanable amount is 10000000.000000002.

The government of the city has the following tax policy:
  - Tax Building by following brackets:
  - Up to 10000 get 3.0% tax
  - Up to 50000 get 8.0% tax
  - Up to 100000 get 13.0% tax
  - Tax Citizen by following brackets:
  - Up to 10000 get 5.0% tax
  - Up to 20000 get 10.0% tax
  - Up to 50000 get 15.0% tax
  - Up to 100000 get 20.0% tax

The government has collected 124774.65254701707 as tax.
  - Tax Citizen are from:
    - 55701.10070718494 are from up to 0
    - 0.0 are from up to 100000
  - Tax Building are from:
    - 0.0 are from up to 100000
    - 69073.55183983216 are from up to 0

The government has distributed 50000 to every citizen as UBI.

There're 200 citizens in the city, and working or living in 122 buildings
    .
During the last month, the city has:
  - Consumption: 489137.5517
  - Investment: 1299054.2543
and total nominal GDP is 1788191.8059999999.
The real GDP, calculated by the base year's price, is 1810789.8210999998.
The GDP deflator is 0.9875203544681556.

The social equality defined as Gini coefficient of income is 0.8564, and
    the wealth inequality is 0.8576.
The unemployment rate is 0.235.
During the past 12 months, the city has:
```

```
 - 1972-02-01 00:00:00: Consumption: 803850.0466, Investment:
    1430793.5346, Government Expense: 50000.0, GDP: 2284643.5812, Wealth
     Gini Coefficient: 0.8619, Income Gini Coefficient: 0.7464,
     Unemployment Rate: 0.335, Homeless Rate: 0.095, Average Happiness:
     0.9475, Average Wage: 1234.325, Average Position Salary: 1755.1287,
     Broad Money Supply: M0 22620930.0, M1 22620930.0, Deposit Rate:
     0.4203, Loan Rate: 0.4421, Reserve Rate: 0.05
[... and other historic information.]

### Interfaces
Here are the actions you may take:
 - adjust_bracket(type: str, brackets: list[tuple[float, float]]): Adjust
    the tax bracket for a given type.
   type: 'Building' or 'Citizen'
   bracket: list of tuples, each tuple contains a threshold and a tax
      rate. 1 means tax 100% and 0.1 means tax 10%.
   e.g. adjust_bracket('Building', [(10000, 0.05), (50000, 0.1),
      (100000, 0.15), (500000, 0.2)])
 - adjust_UBI(amount: float): Adjust the amount of UBI(Universal Basic
    Income) welfare. Noting that if the government has no money, it will
    not distribute UBI.
 - invest(amount: float): Invest government's money into the investment
    market. This action may stimulate the economy.
 - borrow(amount: float): Borrow money from the bank, those money will be
    paid back with interest.
 - payback(loan_id: int): Pay back the loan with the given loan_id.
 - save_money(amount: float): Save money to the bank to get interest.
 - withdraw(amount: float): Withdraw money from the bank.
```

Listing 8: Response example of government agent

```
{
 'strategies': "The city's current balance indicates a significant amount
     of debt with no savings, suggesting that financial management needs
      immediate attention. The high unemployment rate points to economic
     distress, indicating a need for initiatives that stimulate job
     creation and investment. Adjusting tax brackets, especially for
     buildings, could also enhance government revenue, which is crucial
     given the existing loans. Additionally, reviewing the Universal
     Basic Income (UBI) distribution may free up resources for essential
     services.",
 'actions': [
   "adjust_bracket('Building', [(10000, 0.04), (50000, 0.07), (100000,
      0.12)])",
   'invest(10000000)',
   'borrow(10000000)',
   'adjust_UBI(40000)'
 ]
}
```

Listing 9: System prompt example of investment pool

```
You are a city planner in SimCity and you are tasked with planning the
    position and type of a new building.

You must follow the following criteria:
1) The plan should be based on the current city layout and economic
    situation.
2) Act as a city planner and provide a detailed development plan. You may
    construct various buildings and can build multiple instances of the
    same type if deemed suitable.
3) Always refer to the city report, which outlines the current status of
    the city.
4) Determine the type and location of new buildings based on the
    information provided in the city report.
```

```
5) Formulate a step-by-step plan for the construction of the buildings.
6) Factories should be built together and placed away from residential
   areas, while service and commercial buildings should be located near
   residential areas.

There is some information about the city which may affect your plan. The
   information includes two parts: "### Report" and "### Layout"

You should then respond to me with
Reasoning: Are there any steps missing in your plan? What is the purpose
   of each step in the plan? What does the city layout imply? What does
   the city report imply?
x: x_position of the topleft corner
y: y_position of the topleft corner
type: which type of building you'd choose. if you decide not to build,
   type should be "None"

You should only respond in JSON format as described below:
{
    "reasoning": str
    "investments": [
        {
            "x": int,
            "y": int,
            "type": str
        },
        ...
    ]
}
Ensure the response can be parsed by Python `json.loads`, e.g.: no
   trailing commas, no single quotes, etc.

Here are some examples:

INPUT:
### Report
There are several building templates available for construction:
 - Residential Building: Houses 100 people, costs 500 money to build.
 - Factory: Employs 50 people, costs 1000 money to build.
 - Shop: Employs 10 people, generates 200 money per month, costs 500
   money to build.

### Layout
[An Image]

OUTPUT:
{
    "reasoning": "Based on the city layout and report, the city cannot
        produce enough foods. The solution may be build a Restaurant.
        According to the layout, the (25, 25) is a good place.",
    "investments": [
        {
            "x": 25,
            "y": 25,
            "type": "Factory"
        },{
            "x": 23,
            "y": 17,
            "type": "Factory"
        },
        {
            "x": 55,
            "y": 23,
            "type": "Residential Building"
        }
```

```
1458        ]
1459    }
```

Listing 10: User prompt example of investment pool

```
### Report

Here is the average local price of goods in the city:
 - Agriculture_Hunting_Forestry: 121.652
 - Nonenergy_Mining_Products: 35.766
[... and other prices]
 - Health_Social_Work: 168.629
 - Other_Service_Activities: 65.000

If one of the goods is not produced, the price will not be displayed.

There're 200 citizens in the city, and 28 residential buildings have
    capacity, and provide 139 empty houses.
The homeless rate is 0.045, which means 9 citizens are homeless.
and the average rent is 878.4848484848485

Here is the list of buildings that can be built in the city:
 - Agro-Processing Plant(Produces Agriculture_Hunting_Forestry): An agro-
    processing plant that transforms food beverages and agricultural
    products into refined agricultural goods. Cost 50000 to build, needs
     4 employees to run
 - Aquaculture Facility(Produces Fishing_Aquaculture): A facility
    dedicated to aquaculture, producing sustainable seafood and fishery
    products. Cost 50000 to build, needs 5 employees to run
[... and all other templates available]
 - Service Center(Produces Other_Service_Activities): A service center
    that coordinates a variety of professional services and
    administrative support to fulfill community needs. Cost 50000 to
    build, needs 7 employees to run
 - Residential Building: A place where people live. Cost 50000 to build,
    may contain up to 10 people

The size of investment pool is: 3558921.7446145327

Those goods that citizens failed or partially satisfied because of the
    lack of supply or high price:
 - Professional_Scientific_Technical_Activities: Lack 3210.8557714909352
 - Electricity_Gas_Steam_AirConditioning: Lack 1111.6091447969354
[... and all other templates available]
 - Transport_Warehousing_Support: Lack 0.0
 - Household_Employer_Activities: Lack 0.0

### Layout

[An image encoded by base64]
```

Listing 11: Response example of investment pool

```
{
 'reasoning': 'The city report indicates certain deficits in goods and
    services that are not being met, leading to rises in particular
    goods prices and shortages. Several critical sectors are highlighted
     for development: Electricity and Gas, Real Estate,
    Telecommunications, and Food Beverages. Given the current investment
     pool available, it is viable to invest in facilities producing
    these significant needs to reduce shortages and stabilize prices.
    The layout shows ample space available for construction and already
    has a basic infrastructure setup, with residential and other
```

```
      buildings positioned in various grid locations. Key sectors
      identified for investment would benefit from proximity to existing
      buildings for utilities and staff access. Positions (2, 2), (10, 3),
       and (-2, -2) appear to be vacant plots suitable for building based
      on the provided layout image.', 'investments': [
    {'x': 2, 'y': 2, 'type': 'Energy Production Facility'},
    {'x': -2, 'y': -2, 'type': 'Telecommunication Hub'},
    {'x': 10, 'y': 3, 'type': 'RealEstateDevelopmentOffice'},
    {'x': 4, 'y': 2, 'type': 'Food Processing Plant'}
  ]
}
```

Listing 12: System prompt example of firm(Productive Building) agent

```
You are a helpful assistant that utilize the information provided below
    to adjust strategy for my company.

The information of the building will report below "### Report", and
    actions you may do to the building will be listed after "### Actions
    ".
You must follow the following criteria:
1) You should analyze the report and utilize the actions to adjust the
    strategy for the building.
2) You may engage in appropriate equity financing or loan and enhance
    your output by purchasing capital.

You should only respond in JSON format as described below:
{
    "reasoning": "reasoning"
    [
       "action1(para1, para1, ...)",
       "action2(para1, para1, ...)",
       ...
    ]
}
Ensure the response can be parsed by Python 'json.loads', e.g.: no
    trailing commas, no single quotes, etc.

One of the example outputs is as follows:
{
    "reasoning": "The building is in a good condition and the rent is low.
         You should consider increasing the rent to maximize profit.",
    "actions": [
       "IncreaseRent(1000)"
    ]
}
```

Listing 13: User prompt example of firm(Productive Building) agent

```
### Report

Here is the monthly report for Food Processing Plant:
Overall, during this month, the building gross profit is -9028.45, with
    rest cash 0.00. It has 16199.2565 durable investments for promoting
    production.
The building has 2 employees, 2 vacant positions, and run with efficiency
     109.6429%.
The building aims to maximize its profit, and the following is the
    detailed income and expense:

This month, the building income is 17474.907, which consists of the
    following parts:
 - Local trading income: 17474.907(100.0000%)
```

```
This month, the building outcome is 26503.355, which consists of the
    following parts:
 - Other outcome list as follows:

The efficiency of this building is 109.6429%, which consists of the
    following parts:
 - Production Manager has 35.7143% importance, and Kevin is working here.
      Because of Kevin's skills: Management*1.15 Quality Control*1.17 ,
     the efficiency addition is 82.8571%. 1000.0 salary is paid to Kevin.
[... and other employees]

The building has 4 positions, and here are the details:
 - position_id: 2000011, Kevin(Production Manager), salary: 1000.0
[... and other positions]
The unemployment rate in the city is 29.3333%, and the average
    unemployment rate in the past 12 months is 29.3333%.
You may adjust the salary of the position according to the unemployment
    rate.
If the unemployment rate is high, you could open more job positions (in
    practice, converting positions with zero wages into paid positions)
    while reducing the salaries of existing roles to save money.
Alternatively, if the unemployment rate is low, you may adjust the salary
     of the position to a higher level to attract more candidates.
     Additionally, you may want to close certain vacancies that
     consistently fail to attract applicants.

The building run with loss: -9028.4482. And there will be no dividend for
     shareholders.

The social average return on investment is 12.8436%, and the interest
    rate of the bank is 56.0464%.
If roi is higher than the interest rate of the bank, you can invest your
    money in the investment market to get a higher return on investment
    and you may borrow money from the bank to invest if you need.
Otherwise, you may simply save your money in the bank.

The building has the following goods in storage:
 - Food_Beverages: 90356.9299
 - Accommodation_Food_Services: 265.0336

Bank Account Food Processing Plant:
You have no savings
Loans:
 - id: 330, loan amount: 69237.53829366148, should be paid back at
     1972-12 <- DUE!! MUST PAYBACK AS SOON AS POSSIBLE!!!
[... and other loans.]

Demand deposit interest rate is 56.0464%, and loan rate is 58.0693%.
Your loanable amount is 0.

The building has the following price setting for selling goods:
 - Food_Beverages: The average price in the city is 137.4227, changed
     (35.5278%) compared to last year. And the building's price is 200.
[... and other goods.]

There are some items that you may produce more to sell because of
    shortage currently on the market:
 - Food_Beverages: short of 1259.5254643790224 units goods on the market
 [... and other goods.]

During the past 12 months, the building has generated 511752.528 profit,
    running at an average efficiency of 0.588. Capital of the building
    has changed from 7757.525 -> 16199.256 (8441.732).
```

```
 Last year, the building produced 57466.231 goods, and sold
     5929.863(10.3189%) goods. The storage has changed from {'
     Food_Beverages': 50231.5365, '
     Public_Administration_Defence_SocialSecurity': 136.1209, '
     Agriculture_Hunting_Forestry': 109.142} to {'Food_Beverages':
     90356.9299, 'Accommodation_Food_Services': 265.0336}

### Interfaces
Here are the actions you may do to the building:
 - fire(position_id): Terminate the employment of an employee associated
     with a given position_id. This action will result in a one-month
     salary compensation, potentially reducing efficiency but also
     decreasing monthly expenditures
 - adjust_salary(position_id, new_salary): Modify the salary of a
     specific position using its position_id. Increase the salary to
     retain/attract skilled employees, or decrease it to reduce expenses.
 - close_vacancy(position_id): Close a vacancy for a specific position.
     This action will set the salary of the position to 0, effectively
     closing the position temporarily.
 - set_max_running_efficiency(efficiency): Set the maximum running
     efficiency of the building. An efficiency lower than 1 indicates
     that the firm operates below its full capacity (i.e., "idles") to
     reduce production output and material costs.
 - capital_invest(amount): Buying durable goods with [amount] money to
     increase the capital amount of the building. This action will
     improve productivity of the building.
 - equity_finance(amount): Issue shares to raise money. This action will
     increase the current cash of the building.
 - borrow(amount): Borrow money from the bank, those money will be paid
     back with interest. You may borrow money from the bank to invest in
     the building or to cover the cost of the building.
 - payback(loan_id): Pay back the loan with the given loan_id.
 - adjust_price(name, new_price): Alter the price of a particular product
     . If your productivity is in surplus, you may decrease the price to
     sell more goods, and if the product in market is in shortage, you
     may increase the price to boost profit.
 - set_dividend_rate(rate): Modify the dividend rate of the building.
     This rate determines the proportion of profits that will be
     distributed to shareholders. e.g. set_dividend_rate(0.5) will
     distribute 50% of profits to shareholders.
```

Listing 14: Response example of firm(Productive Building) agent

```
{
 'reasoning': 'The building is currently running at a loss and has no
     cash reserves, making it essential to minimize expenses while
     addressing vacancies that could enhance productivity. By reducing
     salaries due to the high unemployment rate and hiring for critical
     roles, particularly the Food Scientist and the vacant Production
     Worker position, profitability can improve. Additionally, immediate
     loan repayments are crucial to avoid further financial penalties.',
 'actions': [
  'close_vacancy(2000012)',
  'close_vacancy(2000014)',
  'adjust_salary(2000011, 800)',
  'adjust_salary(2000013, 500)',
  'borrow(10000)',
  'payback(330)'
 ]
}
```

Listing 15: System prompt example of firm(Residential Building) agent

```
You are a helpful assistant that utilize the information provided below
    to adjust strategy for a residential

The information of the building will report below "### Report", and
    actions you may do to the building will be listed after "### Actions
    ".
You should analyze the report and utilize the actions to adjust the
    strategy for the building.

You should only respond in JSON format as described below:
{
    "reasoning": "reasoning"
    [
        "action1(para1, para1, ...)",
        "action2(para1, para1, ...)",
        ...
    ]
}
Ensure the response can be parsed by Python `json.loads`, e.g.: no
    trailing commas, no single quotes, etc.

One of the example outputs is as follows:
{
    "reasoning": "The building is in a good condition and the rent is low.
         You should consider increasing the rent to maximize profit.",
    "actions": [
        "set_rent(1000)"
    ]
}
```

Listing 16: User prompt example of firm(Residential Building) agent

```
### Report

Here is the monthly report for Residence:
The residential building lives 1 citizens, and its capacity is 10.
The rent per month is 925, and here're other building's rent per month:
 - Residential Building at (14, 5): 900
 - Residential Building at (11, 2): 900
[... and other residential buildings' position and rent]

### Interfaces
Here are the actions you may do to the building:
 - set_rent(new_rent): Alter the rent per month for the residential
    building. This action will directly affect the monthly income of the
     building.
```

Listing 17: Response example of firm(Residential Building) agent

```
{
 'reasoning': 'The building has a low occupancy rate with only 1 out of
     10 units filled. The current rent is at the higher end compared to
     other similar residences nearby. To attract more residents, we
     should consider lowering the rent slightly to improve occupancy.',
 'actions': ['set_rent(900)']
}
```

