# OpenReview forum: "SimCity: Multi-Agent Urban Development Simulation with Rich Interactions"
_ICLR.cc/2026/Conference — ICLR 2026 Conference Withdrawn Submission_

### Official Review · Reviewer_syoF · 2025-10-14

**Soundness:** 3
**Presentation:** 3
**Contribution:** 1
**Rating:** 2
**Confidence:** 5

**Summary:**

This paper presents an ABM with agents using LLM for decision making. The model is designed to simulate the urban development that is similar to the Shellings's segregation model or the NetLogo's urban development model. The agent behavior incurs a series of interactions that result in various optimization from the financial perspective, i.e. taxation. The aggregation of the agent behaviors regenerate the known economic phenomena.

**Strengths:**

I personally am very delighted to see this contribution. From the traditional AI field, ABM has been regarded as a part of AI field, but recently, it has been neglected much. Good integration of LLM and ABM models. Very typical validation process by verifying the regeneration capability of known socio-economic phenomena.

I think that authors would appreciate the above and have done by following the standard ABM research that has been a part of the tradition.

**Weaknesses:**

1.
In spite of my joy, this is not the right venue. AAMAS would be more suitable venue to find your readers. Long ago, AAAI had been such venue, as well, but not much recently. If you consider WinterSim, that would be fine, as well. ACM EC would suppose to accept these topics, but they seldomly do that.

2.
No contribution toward the LLM, itself. A research question that will bring this paper further closer to the ICLR community would be (or NIPS, ICML, etc)
- Does LLM have to be finetuned to fit in your ABM?
- How to lightweight LLM to provide service for massive ABM agents?
- How to draw AI-generated (LLM-generated, or XAI) explanation on a certain behavior from a group of ABM?
- How to stabilize the society faster from external shock with a certain finetuning direction of LLM?

These research questions focus on developing the LLM and its surroundings.

However, the below is your research question from Line 319-323
• RQ1: What phenomena emerge in SimCity, compared with prior simulation environments?
• RQ2: Are emergent regularities robust across multiple simulations?
• RQ3: How does SimCity grow during the move-in phase?
• RQ4: To what extent can SimCity reflect against external shock?

These are questions on simulation aspects, not the LLM used by the agents.
Therefore, the focus is different

3.
Many of agents behavior are still rule based. For instance, a firm model the production by following the Cobb-Douglas function.
I perfectly understand why you model the firm production as such. However, this makes a firm decision-making to be deterministic,
which is not an ideal direction to investigate the agent behavior from the its intelligence standpoint.

4.
It would have been much better to include some fitness analysis to real world datasets. Already, many AAMAS and WinterSim papers
publish such calibration and validation results.

**Questions:**

Please find the weakness section

**Details Of Ethics Concerns:**

No needed because all are hypothetical

---

> ### Author Response · Authors · 2025-11-30
>
> Thank you for your valuable feedback. We have decided to withdraw and further improve it before resubmission.

---

### Official Review · Reviewer_L4Mm · 2025-10-30

**Soundness:** 2
**Presentation:** 2
**Contribution:** 1
**Rating:** 2
**Confidence:** 5

**Summary:**

This paper addresses the challenge of building macro- and urban-scale simulations that both accommodate heterogeneous agents and exhibit canonical economic regularities without hand-crafted rules. It introduces SimCity, a multi-agent framework where households, firms, a central bank, and a government are implemented as LLM agents interacting in labor, goods, and financial markets, with a VLM placing firms on a rendered city map. The system reportedly reproduces patterns such as the Beveridge curve, Okun’s law, Engel’s law, and price elasticity of demand, and shows qualitative robustness across random seeds.

**Strengths:**

1. This paper contrasts DSGE tractability limits and rule-based ABMs with LLM agents that reason in natural language. While not conceptually novel in isolation, the focus on a coherent macro-urban testbed clarifies the intended contribution and evaluation targets.
2. Environment, interaction protocol (function calls with JSON actions), and agent roles, which aid reproducibility and extension. The inclusion of a VLM for geographic firm placement adds a simple path to spatial dynamics without bespoke heuristics.
3. The evaluation compiles a checklist of “stylized facts” and runs cross-framework comparisons (LEN, CATS, EconAgent, AI-Economist/FRED, where applicable). Though the empirical treatment is lightweight, the checklist helps readers situate what phenomena are or aren’t in scope.

**Weaknesses:**

1. Claims of emergent macro relationships are undercut by weak statistical evidence and small samples. For example, the reported Phillips and Okun relationships often have large p-values (e.g., r=-0.14 with p=0.68) over short simulated horizons, which is insufficient to establish the presence, slope, or stability of the curves; visual fit alone is not adequate for ICLR standards. A more rigorous inference protocol (longer horizons, confidence bands, bootstrap across seeds, sensitivity to sampling windows) is needed to substantiate the claims.
2. Novelty relative to existing LLM-agent simulators is incremental and not crisply isolated. The paper contrasts with EconAgent and others, but does not specify which additions (heterogeneous goods, firm templates, VLM siting, central bank rule) are necessary or sufficient for each “stylized fact,” leaving the core methodological advance diffuse rather than sharp. A targeted ablation matrix mapping components to outcomes would clarify unique contributions beyond a larger scenario scope.
3. The economic modeling depth is thin, limiting credibility as a macro platform. Key mechanisms, matching frictions, wage-setting, investment dynamics, and price stickiness, are mostly delegated to prompt-driven LLM behavior, with minimal microfoundations, calibration, or validation against external elasticities; even the Taylor rule employs fixed coefficients without estimation. Without calibration or targets (e.g., impulse responses, variance decompositions), the framework risks being a narrative sandbox rather than a quantitatively informative model.
4. The urban/spatial component is largely cosmetic and weakly evaluated. A VLM “decides” geographic placement and the map renderer shows clustering, but there is no spatial equilibrium, land market, congestion, or transport cost model; the paper presents no quantitative urban metrics (e.g., Moran’s I, monocentric gradients, commuting flows) to support claims about urban expansion dynamics. As a result, the city visuals do not convincingly translate into testable spatial economics.

**Questions:**

Please refer to the weakness.

---

> ### Author Response · Authors · 2025-11-30
>
> Thank you for your valuable feedback. We have decided to withdraw and further improve it before resubmission.

---

### Official Review · Reviewer_326q · 2025-11-02

**Soundness:** 1
**Presentation:** 3
**Contribution:** 2
**Rating:** 2
**Confidence:** 5

**Summary:**

This paper presents SimCity, a multi-agent economic simulation framework using LLMs to model households, firms, a central bank, and government. The system simulates 200 households incorporating 44 heterogeneous goods, labor markets, and financial interactions. A VLM determines placement of buildings on a visualized map. This work evaluates the framework against a checklist of macroeconomic phenomena comparing results with traditional agent-based models.

**Strengths:**

S1. Using LLM-based methods to simulate macroeconomics is a research direction of current community interest with practical significance.

S2. The experiments are conducted based on a series of existing theories of macroeconomic phenomena.

S3. Compared to existing work, this study considers more detailed economic mechanisms and involves a larger number of heterogeneous goods, improving simulation precision. These details are thoroughly described in the appendix.

**Weaknesses:**

W1. There are several flaws in experimental design and interpretation.

W1.1. This work compares results with traditional work (ABM models, etc.). However, SimCity uses far more goods than the baselines, making some selected economic laws incomparable (such as Engel's Law, which the authors mention in line 398 in further tests). Table 2 shows SimCity can only be compared with other work on very few dimensions.

W1.2. Several p-values in the Phillips Curve are not significant. The authors claim that a larger sample size could resolve this issue, but without empirical evidence. SimCity and EconAgent have comparable agent numbers, but EconAgent's results are significant, suggesting the problem may not only lie in sample size as claimed. Okun's Law results also only show that EconAgent has stronger and more significant correlation, which does not match Table 2.

W1.3. Line 365 claims the negative slope can be reproduced with other seeds, but the p-values in Figure 6 also show non-significance. The experimental results contradict the claims. These phenomena cannot explain why SimCity performs worse despite using more complex mechanisms.

W1.4. Line 409 claims the ability to capture elasticity, but this is very likely self-fulfilling behavior based on prompts. As shown in Listing 4 on page 22 (lines 1200-1208), Engel's Law appears to be explicitly modeled to guide agent behavior. Since the prompts already contain economic laws, we cannot determine whether this phenomenon reproduction is spontaneous agent behavior or a result of the prompts.

W1.5. In Table 2, the authors claim to have verified price stickiness, but there seems to be no mention of this anywhere in the paper. Although line 464 mentions this dimension, there is no supporting data evidence.

W1.6. For the volatility in Table 2, only the upper right image in Figure 3 provides results throughout the paper, but there is no explanation for this data. We cannot know how the data was calculated, nor is it clear what the change rate refers to.

W2. RQ3 has limited relevance to the research theme. Urban spatial layout has no obvious relationship with the macroeconomic laws discussed in this work. Additionally, although this work uses VLM to spontaneously form this layout, it does not verify whether this layout approximates actual situations. RQ3 lacks coherence with the other questions.

W3. SimCity appears to be merely an extension of EconAgent and does not provide more contributions at the methodological level of framework design. VLM site selection is also just a simple application of VLM without exploring the significance of this technique within the framework.

**Questions:**

Minor suggestions:

S1. Line 42: the "a" in "agent-based models" should be capitalized.

S2. The right side of Figure 1 showing interaction patterns overlaps with the background, and some fonts are too small, making it difficult to read.

---

> ### Author Response · Authors · 2025-11-30
>
> Thank you for your valuable feedback. We have decided to withdraw and further improve it before resubmission.

---

### Official Review · Reviewer_8xEd · 2025-11-03

**Soundness:** 2
**Presentation:** 2
**Contribution:** 3
**Rating:** 6
**Confidence:** 4

**Summary:**

SimCity introduces a large-language-model-driven ABMs where a mixture of agents act through natural-language reasoning within labor, goods, and financial markets, visualized as a virtual urban map. The paper’s contributions are: (1) integrating LLM agents to enable adaptive, interpretable economic behavior, (2) adding a spatially rendered city via a Vision–Language Model, and (3) validating emergent macro-regularities—such as the Phillips and Okun laws—against canonical stylized facts.

**Strengths:**

I'm really the fan of this work. Especially,

- This paper takes a bold and original step by integrating LLM-based reasoning into agent-based macroeconomic simulation, a domain that has historically relied on fixed rule-based agents.
- The combination of natural-language reasoning, economic interaction, and spatial urban dynamics is genuinely novel and positions the work at the intersection of AI reasoning and computational economics.
- This work opens a promising new direction for LLM-driven social and economic simulation, potentially transforming how researchers study emergent behavior in artificial societies.

**Weaknesses:**

W1. Limited depth and heterogeneity of reasoning.

- Although the agents exhibit more advanced reasoning than traditional rule-based ABMs, the reasoning pipeline remains highly templated and prompt-constrained. As seen in Figure 8, each agent’s behavior follows fixed goal structures and standardized input-output formats. Moreover, heterogeneity among agents is derived only from observable states, while unobservable characteristics (like risk preference on investment) are indirectly inferred from those observables. To improve realism, the reasoning process should model latent attributes explicitly. One constructive direction would be to employ a graphical model where LLMs operate at each node-to-node transition, enabling richer causal reasoning and interdependence among latent and observed variables. As it stands, the system’s “reasoning” is still closer to surface-level text completion than to genuine deliberative cognition.

W2. Lack of micro-level interpretability of emergent phenomena.

- The paper’s analysis of stylized macroeconomic facts is valuable, but the results are presented as static reports rather than explanations. Understanding which specific micro-decisions of the LLM agents led to each stylized fact (e.g., how wage bargaining dynamics produced the Phillips curve) would provide much stronger qualitative evidence of the model’s reliability. Without linking macro patterns to agent-level reasoning traces, it is difficult to assess whether the observed regularities are meaningful or accidental.

W3. Absence of quantitative validation and benchmarking.

- While qualitative reproduction of stylized facts is an excellent start, quantitative alignment with real-world data is essential if SimCity is to be used for policy simulation or empirical analysis. The paper does not evaluate how accurately its LLM-based reasoning reproduces empirical magnitudes compared to rule-based ABMs. For progress toward a scientifically grounded benchmark, the authors should report quantitative fit measures and conduct a direct comparison between traditional ABM reasoning and LLM-powered reasoning under equivalent conditions. In the AI community, incremental improvement is measured through benchmarks; adopting a similar philosophy via automated calibration and reporting measurable performance gains would make the paper much stronger for ICLR.

W4. Scalability bottleneck due to linear LLM calls.

- The current simulation scales linearly with the number of agents, as each agent invokes the LLM independently at every timestep. This linear dependency severely limits scalability. The authors were restricted to fewer than 1000 agents. A promising research direction would be to reduce this cost to sublinear (ideally logarithmic) scaling (possibly via hierarchical aggregation). While such efficiency is beyond the scope of an early-stage study, acknowledging and addressing this scaling issue would position the work toward future large-scale economic simulations.

**Questions:**

-

---

> ### Author Response · Authors · 2025-11-30
>
> Thank you for your valuable feedback. We have decided to withdraw and further improve it before resubmission.

---

### Note · Authors · 2025-11-30

I have read and agree with the venue's withdrawal policy on behalf of myself and my co-authors.